# In Situ Formation of Ti47Cu38Zr7.5Fe2.5Sn2Si1Nb2 Amorphous Coating by Laser Surface Remelting

**DOI:** 10.3390/ma12223660

**Published:** 2019-11-07

**Authors:** Peizhen Li, Lingtao Meng, Shenghai Wang, Kunlun Wang, Qingxuan Sui, Lingyu Liu, Yuying Zhang, Xiaotian Yin, Qingxia Zhang, Li Wang

**Affiliations:** School of Mechanical, Electrical & Information Engineering, Shandong University (Weihai), Weihai 264209, China; peizhenl@yeah.net (P.L.); mlt142536@163.com (L.M.); wkl@sdu.edu.cn (K.W.); qingxuansui@163.com (Q.S.); l1024lly@163.com (L.L.); zyy741790710@163.com (Y.Z.); 15262019723@163.com (X.Y.); 17862703363@163.com (Q.Z.)

**Keywords:** in situ amorphous coating, laser surface remelting, Ti-based alloy

## Abstract

In previous studies, Ti-based bulk metallic glasses (BMGs) free from Ni and Be were developed as promising biomaterials. Corresponding amorphous coatings might have low elastic modulus, remarkable wear resistance, good corrosion resistance, and biocompatibility. However, the amorphous coatings obtained by the common methods (high velocity oxygen fuel, laser cladding, etc.) have cracks, micro-pores, and unfused particles. In this work, a Ti-based Ti47Cu38Zr7.5Fe2.5Sn2Si1Nb2 amorphous coating with a maximum thickness of about 100 μm was obtained by laser surface remelting (LSR). The in-situ formation makes the coating dense and strongly bonded. It exhibited better corrosion resistance than the matrix and its corrosion mechanism was discussed. The effects of LSR on the microstructural evolution of Ti-based prefabricated alloy sheets were investigated. The nano-hardness in the heat affected zone (HAZ) was markedly increased by 51%, meanwhile the elastic modulus of the amorphous coating was decreased by 18%. This demonstrated that LSR could be an effective method to manufacture the high-quality amorphous coating. The in-situ amorphous coating free from Ni and Be had a low modulus, which might be a potential corrosion-resistant biomaterial.

## 1. Introduction

Ti-based alloys are widely used in dental implants and orthopedic prostheses because of their high corrosion resistance and good biocompatibility. However, the elastic modulus of the alloy implant is greater compared with that of bone, causing stress shielding and bone resorption, which ultimately leads to implant failure [1]. To meet the requirements of replacing bones, developing metallic materials with high strength, low elastic modulus, high wear and corrosion resistance, and good biocompatibility is desired [2,3,4]. Ti-based bulk metallic glasses (BMGs) have potential applications in biomedical fields due to their low density, high specific strength, good corrosion resistance, and excellent biocompatibility, etc. [5,6]. However, the application of BMGs is very difficult, for one reason, the preparation of amorphous alloy requires cooling the molten metal liquid at an extremely fast quenching rate (~10^5^ K/s), where the larger the workpiece, the harder to achieve. Otherwise the material will crystallize and lose the remarkable properties of the amorphous. In the Ti-based BMGs, a high content of Be (a toxic element) and Pd (an expensive metal element) are often added to maximize the glass forming ability (GFA), which confines the application of Ti-based BMGs as a biomaterial [6,7]. In this situation, if an amorphous coating can be prepared on the surface of a bulk material, it is possible to obtain significant surface corrosion resistance, wear resistance, and biological adhesion. We no longer consider whether the substrate is completely amorphous, because only the surface properties of the material are considered. Common methods for preparing amorphous coatings are high velocity oxygen fuel (HVOF) [8], laser cladding [9], ion bombardment [10], surface mechanical attrition treatment (SMAT) [11], etc. The coatings obtained by ion bombardment and SMAT are nanoscale, which is easily broken down by abrasion and usually has partial crystallization. The coatings prepared by HVOF and laser cladding are completely amorphous, but have a large number of cracks, micro-pores, and unfused particles, which greatly reduces their immersion corrosion resistance [12]. The in situ formation method might form a coating without these defects. Laser surface remelting (LSR) is accompanied by an instantaneous cooling rate on the order of 10^5^–10^8^ K/s, which is considered to be a more effective method for the surface modification of titanium alloys [13,14]. To date, studies of laser surface modification have usually focused on refining grains and obtaining crystal phases [14,15]. We believe that LSR may be a potential method for preparing amorphous coatings due to its fast cooling rate, ease of changing process parameters, and low cost.

In previous studies, it was found that Ti-based BMGs free from Ni and Be can be formed in a Ti–Cu–Zr–Fe–Sn–Si alloy system. This is considered to be promising for biomedical applications [16,17]. We added a small amount of Nb element to this system to improve the corrosion resistance of the alloy [18]. A high Nb element content will reduce the GFA of the system. Finally, we chose a nominal composition of Ti47Cu38Zr7.5Fe2.5Sn2Si1Nb2 (at. %) as the subject. We intended to develop an amorphous coating with notable corrosion resistance and potential bio-application prospects.

In the present work, LSR was conducted for a prefabricated Ti-based alloy sheet with the laser-induced microstructures and phase changes carefully characterized by back-scattered electron (BSE) imaging and x-ray diffraction (XRD) techniques. In addition, hardness variation across the laser modified zones was examined, and corrosion resistance was also analyzed by anodic polarization experiments.

## 2. Materials and Methods

Alloy ingots with a nominal composition of Ti47Cu38Zr7.5Fe2.5Sn2Si1Nb2 (at. %) were prepared by arc-melting the mixture of the pure elements under a high-purity argon atmosphere. The intermediate alloy was melted five times to ensure the uniformity of ingredients. Plate samples with thicknesses of 2 mm, 1.5 mm, and 1 mm were fabricated by suction casting with water cooled copper molds. The surface of the specimens was polished to 3000-grit by silicon carbide paper and cleaned by absolute ethanol. The laser surface remelting (LSR) experiment was carried out under the protection of a flowing high-purity argon atmosphere, accomplished by a commercial selective laser melting machine (SLM Solutions 125HL, Lubeck, Germany) equipped with a 1067 nm wavelength fiber laser of up to 400 W laser power with a beam diameter of about 80 μm. The surface of the melt receives the effect of spheroidization, causing uneven surface topography (Figure 1a). Figure 1b shows the scheme of the LSR experiment.

The linear energy density (LED), as the single line effective energy input during LSR, can be calculated by P/v (J/mm) [19,20]. P represents the laser power and v represents the scanning speed. In this paper, scanning speed (v) was 2000 mm/s, hatch distance (h) was 140 µm, and laser powers (P) were 0 W, 140 W, 160 W, 180 W, 200 W, 220 W, 240 W, 260 W, 280 W, 300 W, and 320 W. All combinations of different thicknesses of samples and laser powers were processed three times to ensure repeatability. The structures of the treated samples were conducted via x-ray diffraction (XRD, Ultima IV, Rigaku, Tokyo, Japan) with Cu Kα radiation. The nanoindentation characteristics of the sample cross sections were acquired by a Nano Indenter NHT2 (Anton Paar, Austria). The maximum load was 15 mN, and the loading speed was 30 mN/min. Microstructural characteristics under laser treatment were investigated by scanning electron microscopy (SEM, Nova Nano SEM450, FEI Sirion, Hillsboro, OR, USA). To evaluate the corrosion resistance, anodic polarization experiments were implemented in 3.5 wt. % NaCl aqueous solution at room temperature.

## 3. Results and Discussion

XRD patterns of the Ti-based alloy plates treated with different laser powers are shown in Figure 2. All diffraction peaks of the untreated samples with a thickness of 2 mm matched CuTi_2_, CuTi, and Cu_4_Ti_3_ phases. Similar to the treatment with 200 W (LED = 0.10 J/mm), the 240 W (LED = 0.12 J/mm) exhibited a main halo without any Bragg peaks in the XRD pattern, indicating the amorphous structure. The tight combination of the melt and the solid matrix causes rapid heat extraction during solidification, allowing the melt to form a metallic glass under very high cooling rates on the order of 10^5^–10^8^ K/s [14]. For the 280 W, LED = 0.14 J/mm, sharp peaks corresponding to intermetallic compounds CuTi_2_ and CuTi superimposed on the broad peak of amorphous phase were observed (Figure 2a).

The raw samples with a thickness of 1.5 mm had more nanocrystalline structure inside than the 2 mm samples. The nanocrystalline structure was closer to the long-range disorder than the dendrites of the matrix, which had a higher entropy value [21]. From a thermodynamic point of view, the high-conflict atom stacking arrangement corresponded to the “high-entropy” property, and makes the Gibbs free energy drop faster with decreasing temperature, which is conducive to the formation of amorphous structures during solidification. In addition, the laser heat melts the titanium plate during LSR. The generated heat can be dissipated by convection, radiation, and the material itself in a conduction way during remelting [15], while conduction heat loss in a dominant manner (see the blue arrow in Figure 1b). Thick critical dimension heats up more slowly when subjected to the same amount of conduction heat, resulting in a larger temperature gradient, which contributes to achieving a higher cooling rate. Therefore, the crystal in the sample of 1 mm was induced at a lower critical power (160 W, Figure 2b) than the 2 mm sample (280 W, Figure 2a) and the 1.5 mm sample (220 W, Figure 2b). 

Figure 3a is the SEM image of the untreated alloy in backscattered electron (BSE) mode, showing two regions with different morphologies. The surface of the sample was obtained at a faster cooling rate to form a uniform nanocrystal, and the black dendrites were continuously grown due to the slower cooling rate inside. In Figure 3b, the material in the bright region has undergone laser remelting and re-solidification to form an amorphous coating (AC) with a maximum thickness of about 100 μm. There was another modification zone labeled as the heat affected zone (HAZ) beneath the AC. It can be seen that the microstructure of the heat-affected zone is similar to the surface structure in Figure 3a, consisting of uniformly fine nanocrystals. This illustrates the proximity of the HAZ cooling rate to the water-cooled copper mold. After the LSR treatment, the fine and uniform nanocrystalline structure begins to grow under the thermal effect, and grows radially as the dominant form of dark nanocrystals (Figure 3c, inset). Since the nanocrystals have a large atomic diffusion activation energy, the atomic co-diffusion ability is reduced. With the influence of laser heat, the atomic diffusion coefficient increases, and solute redistribution occurs further between dendrites. Ti atoms are enriched in the dark dendritic region, and Zr elements appear in a large number of light regions (Figure 3f). The nanocrystalline structure is gradually consumed as the dendrite grows. Randomly occurring radial dendrites are interconnected (Figure 3c inset), forming directional large-sized dendrite crystals (Figure 3d). The dendrites at the junction grow perpendicular to the interface, consistent with the direction in which the temperature gradient increases (see the red arrow in Figure 3a,b). There were more nanocrystals and few dendrite crystals in the initial state of the 1.5 mm sample (Figure 3e, inset). The temperature gradient of 160 W laser treatment was significantly lower than that of the 200 W sample. The energy density was also relatively low, so the radial dendrites had just been induced to form and cannot continue to grow, which is why it does not have a distinct HAZ region (Figure 3e). It can be seen that the dark dendritic crystal is randomly distributed in the matrix (Figure 3d), with the characteristic size of 100 to 300 µm. In combination with the results of XRD and Energy-dispersive X-ray spectroscopy (EDX), the dark phase was characterized to be primary Cu_4_Ti_3_, and the white regions were Zr-rich, Sn-rich intermetallic compounds with a low melting point.

Figure 4a shows the measured load–displacement curves for the matrix, HAZ, and AC regions in the sample with a thickness of 1.5 mm under 200 W. The corresponding experimental data are given in Table 1. The indentation hardness (HIT) of the matrix, AC, and HAZ are 7995.1, 8514.4, and 12,083 MPa, respectively. The effective elastic modulus in the different regions were obtained based on the obtained indenter’s elastic modulus and Poisson’s ratio, A and S values (Table 1). The effective elastic modulus was 151.78 GPa in the matrix region, 125.55 GPa in the AC, and 153.59 GPa in the HAZ. The intrinsic microstructural origin for the plasticity of amorphous alloys is related to the features of the flow units. In amorphous alloys, the nanoscale liquid-like region (flow units) exhibits a lower atomic bulk density, higher energy states, easier shear deformation, and easier flow than the surrounding area [22,23,24]. The existence of large free volume in amorphous alloys increases the interatomic distance, weakening the atomic bonding energy and atomic migration barrier, resulting in a decrease in hardness and elastic modulus [25]. The load–displacement curve of the AC shows a prominent trait of the intermittent plastic deformation, termed serration or serrated flow [26]. It is generally accepted that the deformation mechanism for amorphous alloys is the nucleation and expansion of shear bands. In our load-controlled tests, the operation of a shear band gives rise to a burst of displacement [27]. Since a serrated flow seems to correspond to each shear band, we expect that larger plastic deformation originates in the accumulation of single shear displacements [27,28]. The distribution of effective elastic modulus from the AC to the matrix region in a sample with thickness of 2 mm under 200 W is illustrated in Figure 4b. Nanoindentation experiments were performed three times for the samples to ensure repeatability. The distribution of the effective elastic modulus in different regions showed the same trend. Furthermore, the effective elastic modulus in the AC zone (123.22 ± 4.64 GPa) was clearly lower than that in the matrix region (149.03 ± 5.76 GPa) and HAZ (153.22 ± 3.96 GPa). The nano-hardness of the 2 mm samples treated by 200 W in the HAZ (10.035 ± 1.45 GPa) was higher than that in the matrix region (8.381 ± 0.279 GPa) and AC zone (8.082 ± 0.396 GPa). Microscopic examination revealed that nanocrystalline dispersion would provide effective precipitation strengthening. Since the structure of nanocrystalline has a short average interatomic distance and strong atomic bonding due to the annihilation of free volume, it has the highest hardness and modulus of elasticity [29].

The anodic polarization curve for the in situ Ti-based amorphous coating in a 3.5% NaCl solution at room temperature in open air, and the curve for specimens deposited under different laser powers are also presented for comparison, as shown in Figure 5. All the anodic portions of the polarization curves exhibit atypical passivation behavior. The corrosion current density (I_corr_) values of all the specimens are shown in the inset image of Figure 5. The I_corr_ value of 240 W was 1.78 × 10^−6^ A/cm^2^, which was about 15% lower than that of the untreated sample (2.12 × 10^−6^ A/cm^2^). Furthermore, the I_corr_ value of 200 W was 1.80 × 10^−6^ A/cm^2^, which was about 27% lower than that of the 320 W (2.45 × 10^−6^ A/cm^2^). According to previous studies, the remarkable corrosion resistance of metallic glasses is contributed by the composition homogeneity with low residual stresses and single-phase nature without grain boundaries and second-phase particles [30,31,32,33]. Ti and Zr are considered to play an important role in corrosion resistance. Zr, Ti, and Nb elements have strong affinity to oxygen, thus, they easily form ZrO_2_, TiO_2_, and Nb_2_O_5_ on the surface of samples after polarization, preventing the direct contact of the corrosive solution with the alloy and thus reducing dissolution of the alloy elements (especially Cu) [34]. Therefore, the existence of a large amount of Ti and Zr in the material leads to a decrease in corrosion current density. However, the Cu element in the Cu-rich region reacts with chloride ions to form CuCl, which subsequently hydrolyzes to Cu_2_O. It provides a galvanic coupling effect for the rapid local dissolution of the Zr-rich and Ti-rich nanometer active regions on the surface of the sample. These active metals also react with the solution, further inducing the formation of local galvanic cells involving Cu [35]. A similar galvanic coupling effect has also been mentioned in another study [18]. In that paper, the addition of the Nb element induced a dual structure containing Ti/Zr/Nb-rich crystalline dendrites and amorphous matrix phases. Regarding the preferential corrosion of the amorphous matrix derived from the formation of a micro galvanic cell between the amorphous matrix and dendrite phases, the results were attributed to the difference in elemental content. In this work, however, the crystalline structure with elemental segregation formed an amorphous coating under laser induction. The amorphous coating has no elemental segregation zone, which is immune to the local dissolution and typically does not form undesirable heterogeneous galvanic cells. Additionally, the addition of Nb promotes the formation of more oxides on the surface, preventing further corrosion of the material. 

## 4. Conclusions

In this paper, the prefabricated alloy plate (Ti47Cu38Zr7.5Fe2.5Sn2Si1Nb2) was processed by laser surface remelting (LSR) technology to obtain an in situ, strongly binding amorphous coating. The in situ formation method makes the coating free from cracks, micro-pores, and unfused particles. The completely amorphous coatings can be obtained with the line energy density (LED) of 0.1 to 0.12 J/mm in 2 mm samples. The corresponding LED was 0.08 to 0.1 J/mm in 1.5 mm samples. By performing the LSR with P = 200 W, v = 2000 mm/s, two distinct modification zones were obtained. The superior hardness promotion (by 51%) was obtained in the heat affect zone (HAZ), and a marked decrease (by 18%) of the elastic modulus in the amorphous coating zone (AC) was present. After the polarization, the addition of the Nb element to form an oxide prevented further corrosion of the material. The amorphous structure without element segregation leads to the failure of micro galvanic cells formation. In addition, the corrosion current density (I_corr_) of the amorphous coating was about 15% lower than that of the untreated specimen (2.12 × 10^−6^ A/cm^2^). Therefore, the in situ amorphous coating obtained by LSR has relatively remarkable corrosion resistance. The composition free from Ni and Be ensures non-toxicity for biological applications. A low modulus of the coating can be better matched to the bone. This might be helpful for the development of corrosion resistant biomaterials. 

## Figures and Tables

**Figure 1 materials-12-03660-f001:**
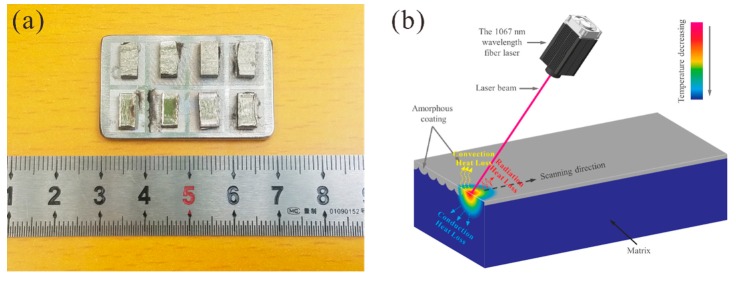
(**a**) The plate samples disposed by different laser powers; (**b**) Scheme of the laser surface remelting.

**Figure 2 materials-12-03660-f002:**
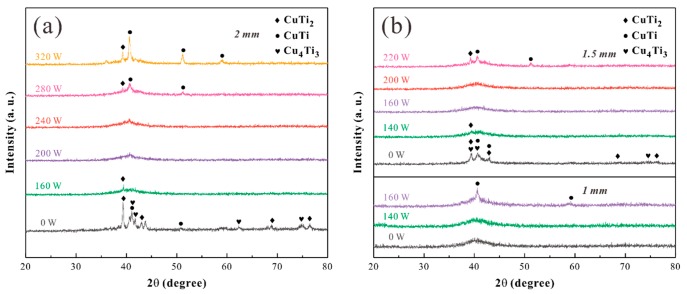
X-ray diffraction (XRD) patterns of samples with (**a**) 2 mm treated under different laser powers; (**b**) 1.5 mm and 1 mm treated under different powers.

**Figure 3 materials-12-03660-f003:**
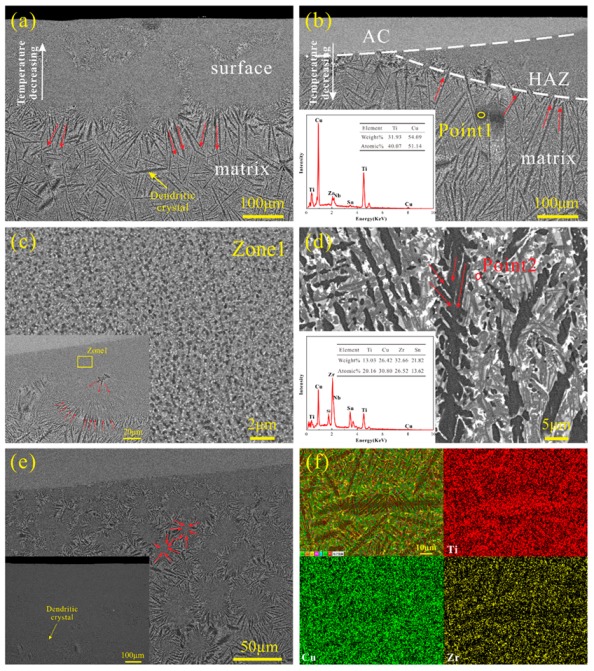
Back-scattered scanning electron microscopy (SEM) micrographs of the (**a**) cross-section of the 2 mm Ti alloy untreated; (**b**) the 2 mm sample under 200 W and the Energy-dispersive X-ray spectroscopy (EDX) corresponding to point 1; (**c**) Heat affected zone (HAZ); and (**d**) matrix regions in the sample with 2 mm under 200 W at high magnification and the EDX of point 2; (**e**) the 1.5 mm sample under 160 W with the inset showing untreated; (**f**) SEM-EDX mapping of dendritic crystals in sample with 2 mm under 200 W.

**Figure 4 materials-12-03660-f004:**
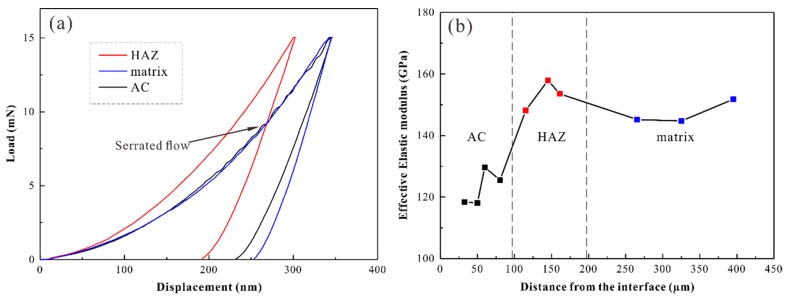
(**a**) Load–displacement curve in different regions of the 1.5 mm sample by 200 W; (**b**) Effective elastic modulus in the different regions near the surface of the 2 mm sample by 200 W.

**Figure 5 materials-12-03660-f005:**
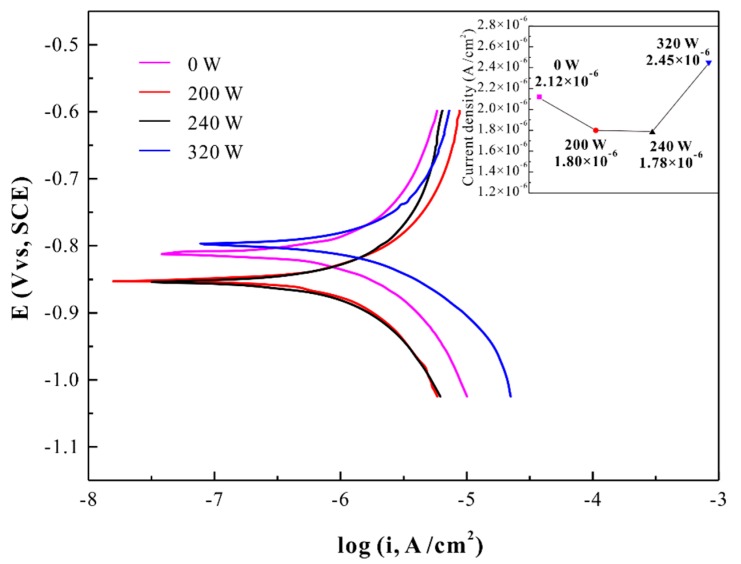
Anodic polarization curves and corrosion current density for the samples with a thickness of 2 mm deposited under different laser powers in a 3.5% NaCl solution.

**Table 1 materials-12-03660-t001:** Nano-hardness and elastic modulus in the different zones of the 1.5 mm sample by 200 W.

Test Parameters	HAZ	Matrix	AC
Maximum load (P_max_, mN)	15	15	15
Indenter’s Poisson’s ratio	0.07	0.07	0.07
Indenter’s elastic modulus (GPa)	1141	1141	1141
Contact depth (h_c_, nm)	231.93	288.88	279.59
Contact area (A, nm^2^)	1,244,136.48	1,878,274.12	1,763,841.78
Contact stiffness (S, mN/nm)	0.1763	0.2143	0.1753
Nano-hardness (H, GPa)	12.083	7.9951	8.5144
Effective Elastic modulus (E*, GPa)	153.59	151.78	125.55

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
