# Peer review of "In Situ Formation of Ti47Cu38Zr7.5Fe2.5Sn2Si1Nb2 Amorphous Coating by Laser Surface Remelting"

_materials, 2019, doi:10.3390/ma12223660_

Round 1

Reviewer 1 Report

The paper brings the results of experimental analysis of the structural, mechanical and corossion properties of the amorphous surface layer created by LSR technology. The topic of this study is interesting, but some changes of the manuscript are needed:

The results of the nano-hardness and effective elastic modulus measurement for 1.5 mm thick sample treated by laser power of 200 W are presented in the Table 1. In the Fig. 4 b is depicted the nano-hardness in different regions for 2 mm thick sample treated by laser power of 200 W. It should be clearly declared in the figure and the table legends.

The statistical aspect of the measurement is taken into account only in the measurement of the elastic modulus (2 mm thick sample treated by 200 W laser power). It is not taken into account in the results shown in the Table 1.

The sentence in Lines 85 – 86 is not well formulated. It must be corrected! In Line 84 is written: “Power change step interval was 20 W.” It is not correct information since the laser powers used in the experiment were: 140, 160, 200, 220, 240, 280, 320 W.

Reviewer 2 Report

Now, in the resubmitted manuscript, dear authors, you tried to highlight what is new, what is your contribution and where the results of your investigation can be applied. 

A few notes about English language. I am not a native English speaker as well as authors (I suppose), but I think that in a newly added sentences, again there are some errors (“However, the amorphous coating obtained by the common methods (high velocity oxygen fuel, laser cladding, etc.) have cracks, micro-pores and unfused particles. … The effects of LSR on the microstructural evolution Ti-based prefabricated alloy sheets were investigated. … The in-situ amorphous coating free from Ni and Be have low modulus, which might be a potential corrosion-resistant biomaterial. … Process each power and thickness combination 85 three times to ensure repeatability. … untreated simples). Please check all added sentences.

The first sentence of the Abstract – Is it a general one or it is about author’s study?

The second sentence of the Abstract – I think that it is a general opinion, but the word “also” is here confusing, because there is no previous statement to compare with.

The sentence “It exhibits better corrosion resistance and its corrosion mechanism was discussed.“ – Better than what?

What is „a minute quantity“? Maybe, some other term could be used?

VERY IMPORTANT: The authors of the present manuscript cited the paper 18. In that paper, the authors investigated the influence of Nb on the properties of the Ti bulk glasses. It would be interesting to compare the results of the present investigation with the results obtained in the cited paper (maybe in the section Results and Discussion). What are the differences? Furthermore, since the importance of Nb was mentioned for the first time in this third version, a few notes about addition of this element should be inserted in Conclusion.

„XRD patterns of the Ti-based alloy plates treated with different laser powers are shown in Fig. 2.“ instead „…in Fig. 2(a).“.

When discussing 2 mm thickness, the authors should mention Fig. 2(a). The similar is valid for 1.5 and 1 mm (Fig. 2(b)).

The text under the Table 1 is stick to the Table.

Finally, my mission (like all reviewers) was to improve the quality of the presentation of the results of your valuable investigation. I believe I have succeeded in this.

Author Response

This manuscript is a resubmission of an earlier submission. The following is a list of the peer review reports and author responses from that submission.

Round 1

Reviewer 1 Report

This submission deals with the investigation of Ti47Cu38Zr7.5Fe2.5Sn2Si1Nb2 amorphous coating obtained by laser surface remelting.

Dear authors, my comments and suggestions are as follows:

GENERAL

VERY IMPORTANT: The conclusion section should highlight the scientific value added of the present manuscript (What is new?; What is the contribution?). In addition, the industrial applicability of findings/results should be underscored (Why the research was conducted?). Please, do not avoid answering these questions.

In many cases, it is unclear for which combination of sample thickness and laser power, the results and discussion are given.

Please, check the English language (a expensive; The in-situ amorphous coating obtained by LSR have better; Therefore, the in-situ amorphous coating obtained by LSR have better; Fig. 2(a) is the SEM image of untreated alloy in a backscattered electron (BSE) mode, shows two regions with different morphologies; leads to decreased the corrosion current density, etc.).

Please, avoid sticking some characters; there are so many errors Nb2(at.%), 1.5mm, 1mm, 15mN, 320watt, highest(280watt, etc. Be also careful with figures.

There are very small letters (invisible) in many figures.

Please, do not use (repeat) very similar sentences (INTRODUCTION - Evaluation of microstructure, nano-hardness and corrosion resistance of treated alloys might be helpful for solving the problem of limited size of amorphous composites. CONCLUSION - The investigation of in-situ amorphous coating in the present study might be helpful for solving the problem of limited application of amorphous composites.). Please, explain the meanings of „limited size“ and „limited application“.

It is common to mention the table and figure in the text first and then insert the table or figure.

Maybe it should be written, for example, Cu2O, (number in subscript), etc.

ABSTRACT

Maybe the word „Ti-based“ can be inserted to the first sentence: The effects of laser surface remelting (LSR) on the microstructural evolution and corrosion resistance of Ti47Cu38Zr7.5Fe2.5Sn2Si1Nb2(at.%) prefabricated alloy sheets were investigated.“

Abbreviations should be written at the first appearance (amorphous coating). 

INTRODUCTION

There are some unclear sentences: „Critical dimension, a core factor, limiting the application of Ti-based BMG systems with low GFA. The amorphous coating with uniform distribution can obtain surface corrosion resistance, hardness and wear resistance comparable to that of metallic glass, while avoiding the demanding preparation requirements of large-sized metallic glass.“

MATERIALS AND METHODS

Are some words missing in the following sentences? „The Linear energy density (LED), the effective energy input per line during LSR, calculated as P/v (J/mm) [14-15]. Here, scanning speeds v=2000 mm/ s, laser power P=140-320watt, hatch distance h=140 μm.“

What is the total number of combinations of different thicknesses of samples and laser powers, i.e. what is the total number of samples used (according to Figure 1c – eleven?; according to figure 1a - eight?)?

RESULTS AND DISCUSSION

The chapter should not be started with figure.

1(c) All?

Please, avoid the term „intimate contact“.

The authors should use uniformly the term „matrix“ or „substrate“ or “base”.

 „see the blue line in Fig. 1b – Where is the blue line?

It seems that the following sentence is very important, but is unclear (because of moderate English language): „Thick critical dimensions create sufficient material matrices that can be dissipated by conduction, which contribute to achieve a higher cooling rate.“ Please, explain in a clearer way.

Figure 3b – GPa instead of Gpa.

of Fig. 4 The Icorr?

REFERENCES

The list of reviewed papers should be organized uniformly (please read the Instructions for Authors) – for example, Inoue, A.; M. Chen; R. RAICHEFF, etc.

Please, check the references 26 and 27 (it is the same paper). The citing of the reference 27 in the text should be checked as well.

Author Response

Response to Reviewer 1 Comments

Dear reviewer, my replies to the suggestions are as follows:

Point 1: GENERAL

VERY IMPORTANT: The conclusion section should highlight the scientific value added of the present manuscript (What is new?; What is the contribution?). In addition, the industrial applicability of findings/results should be underscored (Why the research was conducted?). Please, do not avoid answering these questions.

Response 1: The corresponding modifications marked with a yellow shading have been reflected in the latest manuscript.

For the first time, we have confirmed that LSR is an effective novel method for obtaining in-situ amorphous coatings. Surface modification can be carried out by this method to improve corrosion resistance and mechanical properties. In theory, amorphous coatings of infinite size can be manufactured due to the flexibility of the laser processing technology. (For example, a large-volume work piece can be produced by casting, and then LSR technology can be used to obtain continuous in-situ amorphous coatings that cannot be produced by conventional processes.)

Point 2: In many cases, it is unclear for which combination of sample thickness and laser power, the results and discussion are given.

Response 2: The corresponding modifications marked with a yellow shading have been reflected in the latest manuscript. Please note the explanatory title and caption below the relevant figures.

Point 3: Please, check the English language (a expensive; The in-situ amorphous coating obtained by LSR have better; Therefore, the in-situ amorphous coating obtained by LSR have better; Fig. 2(a) is the SEM image of untreated alloy in a backscattered electron (BSE) mode, shows two regions with different morphologies; leads to decreased the corrosion current density, etc.).

Response 3: The corresponding modifications marked with a yellow shading have been reflected in the latest manuscript.

Point 4: Please, avoid sticking some characters; there are so many errors Nb2(at.%), 1.5mm, 1mm, 15mN, 320watt, highest(280watt, etc. Be also careful with figures.

Response 4: The corresponding modifications marked with a yellow shading have been reflected in the latest manuscript.

Point 5: There are very small letters (invisible) in many figures.

Response 5: The corresponding modifications have been reflected in the latest manuscript. Some of the marks and letters have been enlarged.

Point 6: Please, do not use (repeat) very similar sentences (INTRODUCTION - Evaluation of microstructure, nano-hardness and corrosion resistance of treated alloys might be helpful for solving the problem of limited size of amorphous composites. CONCLUSION - The investigation of in-situ amorphous coating in the present study might be helpful for solving the problem of limited application of amorphous composites.). Please, explain the meanings of „limited size“ and „limited application“.

Response 6: The corresponding modifications marked with a yellow shading have been reflected in the latest manuscript.

However, most Ti-based glassy alloys possess low glass-forming ability (GFA), and the Ti-based BMGs with critical diameters above 5 mm usually contain highly toxic elements such as Be or Ni. The preparation of large size amorphous alloys requires harsh thermal extraction rates. At present, most amorphous alloys are difficult to obtain centimeter-level samples, which limits their engineering applications.

Point 7: It is common to mention the table and figure in the text first and then insert the table or figure.

Response 7: The corresponding modifications marked with a yellow shading have been reflected in the latest manuscript.

Point 8: Maybe it should be written, for example, Cu2O, (number in subscript), etc.

Response 8: The corresponding modifications marked with a yellow shading have been reflected in the latest manuscript.

Point 9: ABSTRACT

Maybe the word „Ti-based“ can be inserted to the first sentence: The effects of laser surface remelting (LSR) on the microstructural evolution and corrosion resistance of Ti47Cu38Zr7.5Fe2.5Sn2Si1Nb2(at.%) prefabricated alloy sheets were investigated.“

Abbreviations should be written at the first appearance (amorphous coating).

Response 9: The corresponding modifications marked with a yellow shading have been reflected in the latest manuscript.

Point 10: INTRODUCTION

There are some unclear sentences: „Critical dimension, a core factor, limiting the application of Ti-based BMG systems with low GFA. The amorphous coating with uniform distribution can obtain surface corrosion resistance, hardness and wear resistance comparable to that of metallic glass, while avoiding the demanding preparation requirements of large-sized metallic glass.“

Response 10: Amorphous alloys have good corrosion resistance and high hardness. However, the preparation of amorphous alloys requires a faster rate of thermal extraction. It is currently difficult to prepare centimeter-sized amorphous alloys. At the same time, amorphous coatings can replace completely amorphous alloys to achieve excellent surface properties.

Point 11: MATERIALS AND METHODS

Are some words missing in the following sentences? „The Linear energy density (LED), the effective energy input per line during LSR, calculated as P/v (J/mm) [14-15]. Here, scanning speeds v=2000 mm/ s, laser power P=140-320watt, hatch distance h=140 μm.“

What is the total number of combinations of different thicknesses of samples and laser powers, i.e. what is the total number of samples used (according to Figure 1c – eleven?; according to figure 1a - eight?)?

Response 11: The corresponding modifications marked with a yellow shading have been reflected in the latest manuscript.

Point 12: RESULTS AND DISCUSSION

The chapter should not be started with figure.

Response 12: The corresponding modifications marked with a yellow shading have been reflected in the latest manuscript.

Point 13: 1(c) All?

Response 13: The XRD patterns does not contain all the results. We have removed similar curves at different powers because the results in the figure are sufficient to illustrate the problem we are trying to discuss.

Point 14: Please, avoid the term „intimate contact“.

Response 14: The corresponding modifications marked with a yellow shading have been reflected in the latest manuscript.

Point 15: The authors should use uniformly the term „matrix“ or „substrate“ or “base”.

Response 15: The word "matrix" has been used uniformly.

Point 16: „see the blue line in Fig. 1b – Where is the blue line?

Response 16: Accurate description should be "blue arrow". The corresponding modifications marked with a yellow shading have been reflected in the latest manuscript.

Point 17: It seems that the following sentence is very important, but is unclear (because of moderate English language): „Thick critical dimensions create sufficient material matrices that can be dissipated by conduction, which contribute to achieve a higher cooling rate.“ Please, explain in a clearer way.

Response 17: Thick critical dimension heats up more slowly when subjected to the same amount of conduction heat, resulting in a larger temperature gradient, which contribute to achieve a higher cooling rate.

Point 18: Figure 3b – GPa instead of Gpa.

Response 18: The corresponding modifications have been reflected in the latest manuscript.

Point 19: The Icorr?

Response 19: The Icorr is corrosion current density.

Point 20: REFERENCES

The list of reviewed papers should be organized uniformly (please read the Instructions for Authors) – for example, Inoue, A.; M. Chen; R. RAICHEFF, etc.

Response 20: The corresponding modifications marked with a yellow shading have been reflected in the latest manuscript.

Point 21: Please, check the references 26 and 27 (it is the same paper). The citing of the reference 27 in the text should be checked as well.

Response 21: The corresponding modifications marked with a yellow shading have been reflected in the latest manuscript.

Please download the attachment to view the latest manuscript.

Thank you for your review.

Kind regards,

Mr. Li

10 Oct 2019

Reviewer 2 Report

The analysis was generally well performed and the discussion of the results was appropriate except for the following issues. Authors said “The metallic glasses provide high corrosion resistance, high strength and excellent wear resistance in comparison with corresponding crystalline alloys” in the introduction. However, nanoindentation test results show that the effective elastic modulus of AC is lower than that of matrix and the hardness of AC is relatively low. Then authors explained the reason by saying “The existence of large free volume in amorphous alloys increases the interatomic distance, weakening the atomic bonding energy, resulting in a decrease in hardness and elastic modulus” Apparently, more detailed explanation is needed to explain why the amorphous phase has a lower modulus than the crystalline phase. It seems necessary to investigate more literature information related to amorphous alloys similar to the Ti alloys in this study.

Author Response

Response to Reviewer 2 Comments

Dear reviewer, my replies to the suggestions are as follows:

Point 1: The analysis was generally well performed and the discussion of the results was appropriate except for the following issues. Authors said “The metallic glasses provide high corrosion resistance, high strength and excellent wear resistance in comparison with corresponding crystalline alloys” in the introduction. However, nanoindentation test results show that the effective elastic modulus of AC is lower than that of matrix and the hardness of AC is relatively low. Then authors explained the reason by saying “The existence of large free volume in amorphous alloys increases the interatomic distance, weakening the atomic bonding energy, resulting in a decrease in hardness and elastic modulus” Apparently, more detailed explanation is needed to explain why the amorphous phase has a lower modulus than the crystalline phase. It seems necessary to investigate more literature information related to amorphous alloys similar to the Ti alloys in this study.

Response 1: Unlike crystalline materials, the atomic packing inside metallic glasses lacks the translational order. Consequently, it is difficult to directly observe the atomic arrangement and its evolution via either crystallography-based microscopy or diffraction methods. Therefore, the atomic structure-property relationship is still missing although substantial progress has been made in the field of metallic glasses. In recent years, it has been found through experiments and computer simulations that there is the nano-scale liquid-like region in amorphous alloys. Compared with the surrounding area, the liquid-like region exhibits lower atomic bulk density, lower hardness and modulus, higher energy state, easy shear deformation and easy flow. Combining these findings, the Chinese Academy of Sciences Institute of Physics/Beijing National Laboratory for Condensed Matter Physics (Weihua Wang) research group proposed a flow unit model to understand and explain the physical and mechanical problems of amorphous materials [1-3]. The free volume in amorphous corresponds to the concept of flow units [4]. We did not specifically analyse the flow unit model and the intrinsic causes of low elastic modulus because this is not the focus of this paper. In the latest manuscript, we cited three articles to explain the relevant issues, and the corresponding yellow shaded mark modifications have been reflected.

References

S. Huo, J.F. Zeng, W.H. Wang, C.T. Liu, Y. Yang, The dependence of shear modulus on dynamic relaxation and evolution of local structural heterogeneity in a metallic glass, Acta Mater. 2013, 61, 4329-4338. H. Wang, The elastic properties, elastic models and elastic perspectives of metallic glasses, Prog. Mater. Sci., 2012, 57, 487-656. Lu, W. Jiao, W.H. Wang, Flow unit perspective on room temperature homogeneous plastic deformation in metallic glasses, H.Y. Bai, Phys. Rev. Lett. 2014, 113, 045501. Turnbull, M.H. Cohen, On the Free‐Volume Model of the Liquid‐Glass Transition, J. Chem. Phys. 1970, 52, 3038-3041.

Thank you for your review.

Kind regards,

Mr. Li

10 Oct 2019

Reviewer 3 Report

The paper brings the results of experimental analysis of the structural, mechanical and corossion properties of the amorphous surface layer created by LSR technology. The topic of this study is interesting, but some changes of the manuscript are needed:

The statistical aspects of the nano-hardness and elastic modules measurement must be added. What were the replications and repetitions of the measurement? What is the variability of the measured data? The sentences in Lines 26 – 27, Lines 49 – 50, Line 69, Lines 93 – 94 and Lines 134 – 135 are not well formulated or contain typographical errors. They must be corrected! Lines 54 and 66: not “the schematic” but “the schematic picture” or “the scheme”. The Fig. 1a and Fig. 1b belong to Chapter 2 (Materials and Methods). The Chapter 3 (Results and discussion) should not start by picture. Line 57: not “P=140-320watt” but “P = 140 – 320 W”. Please use physical units symbols according to SI. Also Line 70, 169. The Fig.1b, Fig. 1c, Fig. 2 and Fig. 4 are not well readable. The must be improved!

Author Response

Response to Reviewer 3 Comments

Dear reviewer, my replies to the suggestions are as follows:

Point 1: The statistical aspects of the nano-hardness and elastic modules measurement must be added. What were the replications and repetitions of the measurement? What is the variability of the measured data?

Response 1: There is no guarantee that the nano-indentation experiments can be performed repeatedly at exactly the same distance, since the test points are manually selected. Therefore, the variability of the measured data cannot be reflected in Figure 4b. However, the mean and standard deviation of the effective elastic modulus in the different regions are given in the latest manuscript. Three times of nano-indentation experiments were performed for the samples of 2 mm under 200 W to ensure repeatability. The distribution of the effective elastic modulus in different regions shows the same trend.

Point 2: The sentences in Lines 26 – 27, Lines 49 – 50, Line 69, Lines 93 – 94 and Lines 134 – 135 are not well formulated or contain typographical errors. They must be corrected! Lines 54 and 66: not “the schematic” but “the schematic picture” or “the scheme”. The Fig. 1a and Fig. 1b belong to Chapter 2 (Materials and Methods). The Chapter 3 (Results and discussion) should not start by picture. Line 57: not “P=140-320watt” but “P = 140 – 320 W”. Please use physical units symbols according to SI. Also Line 70, 169. The Fig.1b, Fig. 1c, Fig. 2 and Fig. 4 are not well readable. The must be improved!

Response 2: The corresponding modifications marked with a yellow shading have been reflected in the latest manuscript. Some of the marks and letters in the figures have been enlarged.

Thank you for your review.

Kind regards,

Mr. Li

10 Oct 2019

Round 2

Reviewer 1 Report

The authors did some effort to make improvements in the manuscript. Unfortunately, many errors remained. The effort done should be a much better for a Q1 journal.

Still, many sentences are unclear (English language should be improved). These are:

“The Linear energy density (LED), the effective energy input per line during LSR, calculated as P/v (J/mm) [14-15].

Here, scanning speeds v=2000 mm/ s, hatch distance h=140 μm, laser power P=140-320 W.

Fig. 4(b) shows that the distribution of effective elastic modulus from the AC to the matrix region in sample with thickness of 2 mm under 200 W.

Many literatures have reported that the composition homogeneity with low residual stresses and single-phase nature without grain boundaries and second-phase particles of the metallic glasses, which could be  responsible for the remarkable corrosion resistance [25-28].

The in-situ amorphous coating obtained by LSR has better corrosion resistance, as a lower corrosion current density (Icorr) about 15% than that of the untreated specimen (2.12E-6A/cm2).”

Furthermore, the following question is not answered:

“What is the total number of combinations of different thicknesses of samples and laser powers, i.e. what is the total number of samples used (according to Figure 1c – eleven?; according to figure 1a - eight?)?”

In addition, only mentioned examples to change were changed, but the similar ones were not (CuTi2, Cu4Ti3, 0.10J/mm, 0.12J/mm, 0.14J/mm, CuTi2, Intensity(u. a.), 2q(degree), and153.59, Fig. 5 The, 6A/cm2 , 6A/cm2 , 6A/cm2,  2000mm/s, 6A/cm, Yeh, J. , Chen, S. , Lin, S. , Gan, J. , Chin, T. , Shun, T. , Tsau, C. and Chang, S.,).

Author Response

Dear reviewer, my replies to the suggestions are as follows:

Point 1: Still, many sentences are unclear (English language should be improved). These are:

“The Linear energy density (LED), the effective energy input per line during LSR, calculated as P/v (J/mm) [14-15].

Here, scanning speeds v=2000 mm/ s, hatch distance h=140 μm, laser power P=140-320 W.

Fig. 4(b) shows that the distribution of effective elastic modulus from the AC to the matrix region in sample with thickness of 2 mm under 200 W.

Many literatures have reported that the composition homogeneity with low residual stresses and single-phase nature without grain boundaries and second-phase particles of the metallic glasses, which could be responsible for the remarkable corrosion resistance [25-28].

The in-situ amorphous coating obtained by LSR has better corrosion resistance, as a lower corrosion current density (Icorr) about 15% than that of the untreated specimen (2.12E-6A/cm2).”

Response 1:

The revised sentence is as follows:

The Linear energy density (LED), as the single line effective energy input during LSR, can be calculated by P/v (J/mm) [14-15]. (Line 55-56)

In this paper, scanning speed (v) was 2000 mm/s, hatch distance (h) was 140 μm and laser powers (P) were 140 to 320 W. (Line 57-58)

The distribution of effective elastic modulus from the AC to the matrix region in a sample with thickness of 2 mm under 200 W is illustrated in Fig. 4(b). (Line 142-144)

According to previous literatures, the remarkable corrosion resistance of metallic glasses is contributed by the composition homogeneity with low residual stresses and single-phase nature without grain boundaries and second-phase particles [25-28]. (Line 162-165)

The in-situ amorphous coating obtained by LSR has possess relatively terrific corrosion resistance. In addition, its corrosion current density (Icorr) is about 15% lower than that of the untreated specimen (2.12E-6 A/cm2). (Line 187-189)

(The corresponding modifications marked with a yellow shading have been reflected in the latest manuscript.)

Point 2: Furthermore, the following question is not answered:

“What is the total number of combinations of different thicknesses of samples and laser powers, i.e. what is the total number of samples used (according to Figure 1c – eleven?; according to figure 1a - eight?)?”

In addition, only mentioned examples to change were changed, but the similar ones were not (CuTi2, Cu4Ti3, 0.10J/mm, 0.12J/mm, 0.14J/mm, CuTi2, Intensity(u. a.), 2q(degree), and153.59, Fig. 5 The, 6A/cm2 , 6A/cm2 , 6A/cm2,  2000mm/s, 6A/cm, Yeh, J. , Chen, S. , Lin, S. , Gan, J. , Chin, T. , Shun, T. , Tsau, C. and Chang, S.,).

Response 2:

Laser power (P) was 140 W to 320 W. Power change step interval was 20 W. The total number of combinations of different thicknesses of samples and laser powers was eleven. Process each power and thickness combination three times to ensure repeatability.

The corresponding modifications marked with a yellow shading have been reflected in the latest manuscript. (Line 58-60)

The numbers in subscript have been modified. (line 71, 72, 77, 117) Spaces have been added between numbers and letters. (line 72, 76, 130, 159-161, 182, 187) The list of references has been organized uniformly. (line 233) The contents of Figures 2, 4 and 5 have been modified.

Thank you for your review.

Kind regards,

Mr. Li

14 Oct 2019